# Identifying Genetic Variants and Metabolites Associated with Rapid Estimated Glomerular Filtration Rate Decline in Korea Based on Genome–Metabolomic Integrative Analysis

**DOI:** 10.3390/metabo12111139

**Published:** 2022-11-19

**Authors:** Sangjun Lee, Miyeun Han, Sungji Moon, Kyungsik Kim, Woo Ju An, Hyunjin Ryu, Kook-Hwan Oh, Sue K. Park

**Affiliations:** 1Department of Preventive Medicine, Seoul National University College of Medicine, Seoul 03080, Republic of Korea; 2Cancer Research Institute, Seoul National University College of Medicine, Seoul 03080, Republic of Korea; 3Department of Biomedical Sciences, Seoul National University Graduate School, Seoul 03080, Republic of Korea; 4Department of Internal Medicine, National Medical Center, Seoul 04564, Republic of Korea; 5Interdisciplinary Program in Cancer Biology, College of Medicine, Seoul National University, Seoul 03080, Republic of Korea; 6Integrated Major in Innovative Medical Science, Seoul National University College of Medicine, Seoul 03080, Republic of Korea; 7Department of Internal Medicine, Seoul National University Hospital, Seoul 03080, Republic of Korea

**Keywords:** single-nucleotide polymorphism, kidney function, estimated glomerular filtration rate, genome-wide association study

## Abstract

Identifying the predisposing factors to chronic or end-stage kidney disease is essential to preventing or slowing kidney function decline. Therefore, here, we investigated the genetic variants related to a rapid decline in the estimated glomerular filtration rate (eGFR) (i.e., a loss of >5 mL/min/1.73 m^2^ per year) and verified the relationships between variant-related diseases and metabolic pathway signaling in patients with chronic kidney disease. We conducted a genome-wide association study that included participants with diabetes, hypertension, and rapid eGFR decline from two Korean data sources (N = 115 and 69 for the discovery and the validation cohorts, respectively). We identified a novel susceptibility locus: 4q32.3 (rs10009742 in the *MARCHF1* gene, beta = −3.540, P = 4.11 × 10^−8^). Fine-mapping revealed 19 credible, causal single-nucleotide polymorphisms, including rs10009742. The pimelylcarnitine and octadecenoyl carnitine serum concentrations were associated with rs10009742 (beta = 0.030, P = 7.10 × 10^−5^, false discovery rate (FDR) = 0.01; beta = 0.167, P = 8.11 × 10^−4^, FDR = 0.08). Our results suggest that *MARCHF1* is associated with a rapid eGFR decline in patients with hypertension and diabetes. Furthermore, *MARCHF1* affects the pimelylcarnitine metabolite concentration, which may mediate chronic kidney disease progression by inducing oxidative stress in the endoplasmic reticulum.

## 1. Introduction

Chronic kidney disease (CKD) is a worldwide public health concern [1]. Patients with CKD have an increased risk of end-stage kidney disease (ESKD) and cardiovascular disease. Therefore, identifying the predisposing factors for CKD or ESKD is essential to preventing or slowing the rate of kidney function decline [2,3].

Genetic susceptibility is also a risk factor for CKD, in addition to diabetes mellitus and hypertension [4]; CKD heritability is estimated to be between 30 and 75% [5]. Several genome-wide association studies (GWASs) have identified genetic loci associated with CKD in populations comprising millions [6,7]. The first GWAS published in 2009 identified *UMOD*, *SHROOM3*, and *STC1* to be associated with renal phenotypes such as estimated glomerular filtration rate (eGFR), creatinine and cystatin C, CKD, tubulo-interstitial inflammation, and renal fibrosis [8,9,10]. GWASs are important for mapping the risk loci for complex diseases by identifying the association between genetic variants and diseases [10]. Prior to the development of SGLT2 inhibitors in large-scale clinical trials, conventional therapies used to slow the decline in renal function were only moderately effective on clinically relevant renal endpoints [11,12]. Selecting the genetically supported drugs targeting the causal genes from Mendelian diseases or GWAS-driven coding variants was estimated to double the success rate in drug discovery [13,14]. Moreover, it is possible to discover the candidate genes for drug development by identifying the key genes associated with specific diseases through an in silico functional analysis based on a meta-analysis from published GWASs [15]. These underlie GWASs for the identification of genetic variants associated with the deterioration of renal function.

However, only a few GWASs have explored eGFR decline [16,17,18], despite being a surrogate marker for ESKD [19,20]. The *Kidney Disease: Improving Global Outcomes* guidelines define a rapid progression as a sustained eGFR decline of >5 mL/min/1.73 m^2^ per year [21]. However, no study has directly investigated this.

Asian patients with CKD tend to progress faster to ESKD than other ethnic groups [22]. Since the origin of ESKD risk mismatch between Asians and other ethnic groups is not accounted for by traditional risk factors such as exposure to specific dietary products, socioeconomic status, or comorbid imbalances, the potential roles of nontraditional risk factors are highlighted [23]. Studies have noted that the prevalence of IgA nephropathy is higher among Asians [24,25]. The cultural factors of Asian traditional herbs and other therapies are also potential reasons [26]. Select Asian herbs and remedies may contain poorly defined nephrotoxic compounds [26].

A GWAS using cross-sectional eGFR meta-analysis data from an Asian population has been performed [27], but the authors did not investigate eGFR decline. Therefore, we aimed to identify the genetic variants associated with a rapid eGFR decline in the Korean general population. Furthermore, identifying genetic variants alone may be insufficient. Metabolites are biological pathway end-products. Thus, a recent integrated study evaluated the effects of genetic variants on the phenotypes associated with metabolite enrichment to enhance our understanding of the biological mechanisms and networks [28]. Therefore, we also evaluated the associations between genetic variants and serum metabolite enrichment using a genome–metabolomic integrative analysis (GMIA).

## 2. Material and Methods

### 2.1. Data Sources and the Study Population

The Korean Biobank Array, also called the Korean Chip (K-CHIP) Consortium, consists of three general population cohorts with genomic information: the Health Examinee Cohort Study (HEXA), the Cardiovascular Disease Association Study (i.e., CAVAS), and the Korea Association Resource (KARE). Thus, we used K-CHIP as our discovery dataset for GWASs to explore rapid eGFR decline. The K-CHIP Consortium, designed by the Center for Genome Science, contains approximately 8,000,000 single-nucleotide polymorphisms (SNPs) customized for the Korean population [29]. Details about the quality control and imputation of the K-CHIP Consortium have been described previously [29].

We used a validation dataset from 2045 CKD samples to verify our GWAS results. We collected 2306 samples, comprising 2118 subjects from the KoreaN Cohort Study for Outcomes in Patients with Chronic Kidney Disease (KNOW-CKD) cohort and 188 participants with biopsy-proven diabetic nephropathy from two hospitals (91 patients from the Seoul National University Hospital Human Biobank and 97 from Kyung Hee University Medical Centre). The KNOW-CKD is a multicenter, prospective, observational study of 2388 patients with CKD [30]. In total, 2045 samples passed the genotype quality control process, resulting in 7,763,720 remaining SNPs, similar to the K-CHIP Consortium; these were included in the study [29].

We screened 72,298 participants from the K-CHIP Consortium and 2045 from the CKD cohort for the GWAS analysis. First, we excluded participants with a history of cancer, those with missing eGFR data, and those with an eGFR slope of less than −5 mL/min/1.73 m^2^ per year. Next, we excluded participants without both diabetes and hypertension. Finally, 115 participants were included from the K-CHIP Consortium (discovery cohort), and 69 participants were included from the CKD cohort (validation cohort) (Appendix A). The median follow-up of the K-CHIP consortium was 3.9 (interquartile range [IQR] = [3.6, 4.2]), and that of the KNOW-CKD was 1.8 ([IQR] = [1.4, 2.4]) (Table 1).

We also measured 135 serum metabolites for all 2580 participants in the KARE cohort that is part of the Korean Genome and Epidemiology Study (i.e., KoGES) (Appendix A). The quality control measures for the serum metabolites in the KARE cohort have been previously described [31]. Of the 2580 participants, 1905 had information about both genotypes in the K-CHIP Consortium. Finally, 137 patients with hypertension and diabetes were included.

### 2.2. Exposure Measurements

The eGFR was calculated using the four-variable Chronic Kidney Disease Epidemiology Collaboration equation [32]. For the discovery cohort, only subjects with at least two follow-up eGFR measurements were selected. The eGFR change was calculated by dividing the difference in the eGFR by the follow-up year and using linear mixed models with random intercepts in the validation cohort.

### 2.3. GWAS

We performed a GWAS to evaluate rapidly declining eGFR (i.e., a decline greater than 5 mL/min/1.73 m^2^ per year) using a linear regression analysis under the assumption of an additive genetic model; the study was based on the K-CHIP consortium (the discovery cohort) and was conducted using the Pass kinship analysis (PLINK) version 2.0 (http://pngu.mgh.harvard.edu/∼purcell/plink) [33]. SNPs with a *p*-value below 1 × 10^−6^ were considered to have genome-wide significant associations.

Significant associations from the K-CHIP consortium were validated using data from independent patients with CKD. SNPs with *p*-values of less than 0.05 were considered valid. The annotation for selected SNPs and linkage disequilibrium (LD) clumping (R^2^ < 0.001 within a 10,000 kb window) was conducted from the reference panel of phase 3 in the 1000 Genomes project (East Asian) using the “ieugwasr” R package (R software, version 4.1.2 R Core Team, Vienna, Austria) and the ANNOtate VARiation (ANNOVAR) (version 20220320) [34,35].

### 2.4. Fine-Mapping

Fine-mapping was performed to find the potential causal variants among the SNPs identified in the GWAS [36] using the sum of single effects (SuSiE) method [37]. The lead SNP was determined based on the strongest association signal. We then selected +/−10 kb regions around this SNP. Based on an iterative Bayesian stepwise selection, the SuSiE regression model calculated the posterior inclusion probability (PIP) for each SNP, which is the probability of including the SNP in a causal association, by isolating the effects of the LD structure (1000 Genomes Project East Asian population) [36]. Credible sets were created through an iterative model fitting generated by ranking the SNPs from the largest to the smallest PIP. A regional plot around the lead SNP was created by “LocusZooms” and fine-mapping analysis was conducted using the “susieR” R package [37,38].

### 2.5. GMIA

GMIA was performed using a linear regression model to identify the metabolomic mechanisms underlying the genetic variants. A linear regression model was constructed based on the metabolite’s serum concentration, calculated through metabolomic analysis, and the previously calculated allele information from GWAS analysis based on an additive model. The associations were considered significant if the false discovery rate (FDR) was <0.05.

## 3. Results

Table 1 presents the participant’s general characteristics for the discovery and validation cohorts. The discovery cohort participants were older and had higher baseline hemoglobin, serum albumin, and eGFR levels, and more follow-up years than the validation cohort. Furthermore, they had lower systolic and diastolic blood pressures and body mass indices than those in the validation cohort. However, the eGFR slope and sex did not differ between the cohorts.

Appendix A presents the Manhattan and quantile–quantile plots of GWAS analysis for both cohorts. The genetic inflation was 1.004 and 0.995 in the discovery and the validation cohorts, respectively (Appendix A).

We identified 241 SNPs associated with a rapid eGFR decline in the discovery cohort (Appendix A). Of these, we identified five externally validated SNPs (rs10009742, rs1390835129, rs71600637, rs6852270, and rs5012631) on the membrane-associated ring-CH-type finger 1 (*MARCHF1*) gene significantly associated (genome-wide) with a rapid eGFR decline (Table 2). Of these five SNPs, rs10009742 (discovery cohort: beta = −4.128, standard error [SE] = 0.790, *p*-value = 8.01 × 10^−7^; validation cohort: beta = −2.361, SE = 1.118, *p*-value = 0.04 in the validation cohort; meta-analysis with a fixed-effect model: beta = −3.540, SE = 0.645, *p*-value = 4.11 × 10^−8^) was selected as the lead SNP after LD clumping (Table 2).

SNPs correlating with rs10009742 on LD were identified from a regional plot with a 50 kb interval from rs10009742 to identify the potential causal SNPs by fine-mapping (Figure 1A). After fine-mapping at 10 kb intervals by enlarging the regional plot, we estimated 19 credible sets that could be considered causal SNPs based on rs10009742 with respect to the PIP for the 4q32.3 (164664901–164684901), 70 loci (Figure 1B; Appendix A). Of the 19 credible sets, rs13127646 had the highest estimated PIP.

GMIA identified four genome-wide significant SNPs (rs10009742, rs1390835129, rs71600637, and rs6852270) associated with the pimelylcarnitine (beta = 0.030, SE = 0.007, FDR = 0.01) and octadecenoylcarnitine (beta = 0.167, SE = 0.049, FDR = 0.08) (Table 3) serum concentrations. We also observed an increase in the pimelylcarnitine and octadecenoylcarnitine blood concentrations per the effect allele (T) of rs10009742 (Figure 2).

## 4. Discussion

The main aim of our study was to identify the mechanisms of integration between genetic variants and metabolites based on the association between the genetic variants selected through GWAS analysis and the metabolites that are the last product in biological pathways. We found that the rs10009742 in the *MARCHF1* gene on chromosome 4q32.3 was associated with a rapid eGFR decline (a decrease of ≥5 mL/min/1.73 m^2^ per year) in patients with hypertension and diabetes in Korea. The serum pimelylcarnitine concentration was also associated with the effective allele (T) of rs10009742 compared with the reference genotypes (C/C).

The *MARCHF1* gene has been associated with fatty acids (FAs), glucose metabolism, and renal dysplasia [39,40]. MARCHF1 is a membrane-bound ubiquitin ligase that mediates protein ubiquitination [41]. The ubiquitin (Ub)–proteasome system (UPS) tags and degrades proteins, and there is evidence that Ub and proteasome subunit transcription is involved with UPS-induced muscle proteolysis in CKD. Additionally, previous CKD studies on CKD complications have shown that inflammation and acidosis activate Ub junctions, causing muscle proteolysis in CKD; this suggests that MARCHF1 is linked to the development of CKD [42,43,44].

CKD is caused by the progression of transient acute kidney injury (AKI) of a fully reversible lesion, which can be initiated by secondary causes, such as hypertension and diabetes mellitus [45]. AKI increases the CKD risk in patients with transient AKI, which is accompanied by a fibrotic outcome. A proposed AKI to CKD progression mechanism is fatty acid oxidation (FAO) downregulation in tubular epithelial cells [46]. Oxygen deprivation (a major cause of AKI) can stop FAO, resulting in a long-term decrease in energy supply to cells, namely starvation. Furthermore, FAO downregulation is associated with lipid accumulation in the kidneys and the liver, leading to tubulointerstitial inflammation that contributes to fibrosis [46]. A long-term lack of energy, such as during starvation, prolonged exercise, illness, and fever, triggers endoplasmic reticulum (ER) stress, inducing autophagy and apoptosis. Consequently, FAO inhibition by mitochondrial dysfunction occurs in the kidney, liver, heart, and skeletal muscles (Figure 3) [47,48,49,50].

MARCHF1 ubiquitination modulates tubulo-interstitial inflammation, renal fibrosis, and FAO [44]. Furthermore, it regulates tubular-interstitial inflammation, renal fibrosis (leading to CKD), and muscle protein breakdown in CKD [44]. However, as AKI progresses to CKD, β-oxidation damage predominates over the normal regulatory capacity for ubiquitination, resulting in an increased intracellular accumulation of FAs and CKD exacerbation [51,52]. Previous studies using obesity, diabetes, and starvation animal models support our mechanistic explanation. For example, acylcarnitine (AC) accumulation has been associated with mild FAO dysregulation and mitochondrial stress, exacerbating insulin resistance [48]. Hypertension may also affect FA transport via a cluster of differentiation 36/fatty acid translocase (CD36/FAT). One study suggested that the onset of myocardial metabolic disorders during the early stage of hypertension decreased plasma CD36/FAT content and function, leading to decreased FA transport capacity (Figure 3) [53].

An inherent problem of metabolism is that it interferes with FA catabolism, resulting in a considerable increase in the plasma and urine concentrations of long- and medium-chain fatty acids [54]. A previous study in rodents reported that short-, medium-, and long-chain ACs accumulate in the serum and muscles owing to insulin resistance impairments arising from mismatches among long-chain fatty acid delivery, catabolism, and the tricarboxylic acid cycle rate (Figure 3) [48].

Excessive ubiquitination caused by genetic variants of rs10009742 in *MARCHF1* impairs the activity of cellular insulin by degrading insulin receptor-β on the cell surface, leading to diabetes [55]. In addition, CD36/FAT degradation interrupts the cellular transport of long-chain fatty acids in patients with hypertension and genetic variants of rs10009742 [53]. Subsequently, FA accumulates in the blood, resulting in decreased FAO, which causes ER stress due to β-oxidation damage. Thus, medium-chain fatty acids in plasma cannot diffuse into the cells due to long-chain AC accumulation from ER stress [56,57].

Our study indirectly demonstrates these mechanisms; we detected an increased serum pimelylcarnitine concentration, which is a long-chain AC with an effective allele (T) of rs10009742 on *MARCHF1* (Figure 3). A series of mechanisms related to hypertension and diabetes can exacerbate the development and progression of CKD. Therefore, this suggests that MARCHF1 and its regulatory metabolites are crucial in triggering CKD progression. In addition, octadecenoylcarnitine, a long-chain AC, may also be involved in the kidney, although its association with rs10009742 on *MARCHF1* was not statistically significant based on the FDR value.

Our study has some limitations. First, the definition of eGFR change in the epidemiological data is less clear than that in the clinical data, which generally defines eGFR change as a decline for at least three follow-up visits [19,58]. The HEXA cohort from the K-CHIP Consortium in our discovery dataset had only two follow-ups. Thus, the eGFR change in the HEXA cohort was calculated based on the two time points divided by the follow-up year. Considering this, we attempted to select patients with CKD from the general population dataset and the K-CHIP Consortium, and then selected patients with CKD progression using repeated eGFR data. For this purpose, in this study, we selected subjects based on a very rapid eGFR decline. In addition, we only selected high-risk participants with a rapid eGFR decline, hypertension, and diabetes. Thus, the number of study participants could be insufficient. Therefore, the associations between the SNP and the metabolites were indirectly estimated among other participants in the GWAS. Nevertheless, compared to the general population, high-risk participants can be more appropriate for determining the effects of genetic variants [59]. However, validation analysis is required for a large consortium study in the future. Furthermore, only serum metabolite concentrations associated with diabetes were considered in our study [31], but urine metabolites are potential renal biomarkers [60]. Since diet and gut microbiome composition are likely associated with the metabolite profile of various diseases, multi-omics designs that include various metabolites are required to completely understand the metabolomic mechanisms.

In cohort studies, selection bias that can affect either the internal or the external validity of a study occurs when the selection of exposed and nonexposed participants is associated with the outcomes [61,62]. Therefore, selection bias can also occur if a subgroup of participants (such as a high-risk group or a more disease-susceptible group) within a study is selected for more detailed research [63]. Nevertheless, the problem related to selection bias may not be significant because the genetic variants that are a factor at birth can be free from traditional confounding factors [64,65,66]. Therefore, the problems associated with a lack of representativeness that can cause selection bias may be modest [64,65].

Furthermore, we identified genetic variants in a population more susceptible to CKD development to account for the missing heritability. Schematically, studies to account for missing heritability involve genomic analysis on a high-risk population to identify major genes [67]. The prevalence of renal dysfunction in patients with hypertension and diabetes is higher than that in patients with hypertension or diabetes alone [68]. Hypertension and diabetes have common metabolic pathways, and genetic, environmental, and behavioral factors contribute to the comorbidity of the two diseases [69]. Therefore, a high-risk group was studied to exclude missing heritability that may occur owing to environmental influences in estimating the genetic effect on CKD development based on the general population.

Nonetheless, the major strength of our study is the GWAS, which was performed in patients with hypertension and diabetes from the general population and validated in a population of patients with CKD in Korea. Genetic variants of *MARCHF1* may be associated with the development and progression of CKD in the general population. Therefore, MARCHF1 could be clinically significant since it has the potential to interfere with the development and progression of CKD. Therefore, this result provides an opportunity for novel drugs targeting MARCHF1. A previous study analyzed the association between metabolites and CKD [70]. However, the genetic effects on the metabolite concentrations in our study could be clinically significant because circulating metabolites have broad effects on renal function. Thus, these metabolites possibly have functional roles in the development and progression of CKD.

In conclusion, the *MARCHF1* gene on chromosome locus 4q32.3 (rs10009742, reference/effective allele, C/T) was associated with a very rapid decline in eGFR, an indicator of CKD progression, in patients with hypertension and diabetes in Korea. Furthermore, the rs10009742 genetic variation in *MARCHF1* can be modified by the serum pimelylcarnitine concentration. Overall, our study provides insight into interventions for patients with hypertension and diabetes in Korea at high risk for CKD development and progression by estimating the effects of genetic variants on the metabolites within circulating metabolic mechanisms.

## Figures and Tables

**Figure 1 metabolites-12-01139-f001:**
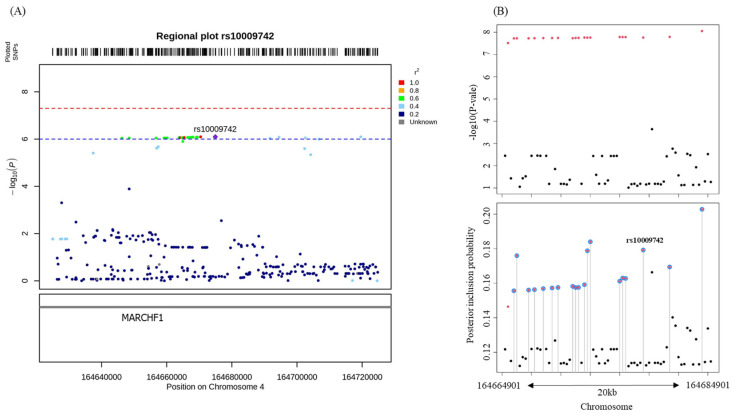
Genomic characteristics for the rs10009742 in 4q32.3. (**A**) Regional plot for the regions around rs10009742 in 4q32.3. The red point indicates the most strongly associated lead SNP, rs10009742 (purple diamond). (**B**) Fine-mapping results for rs10009742, illustrated by the PIP plot of fine-mapping around rs10009742 with +/−10 kb intervals. The SNPs around the blue circles in the plot are credible sets with a potential causality (the highest PIP was estimated for rs13127646). PIP, posterior inclusion probability; SNP, single-nucleotide polymorphism.

**Figure 2 metabolites-12-01139-f002:**
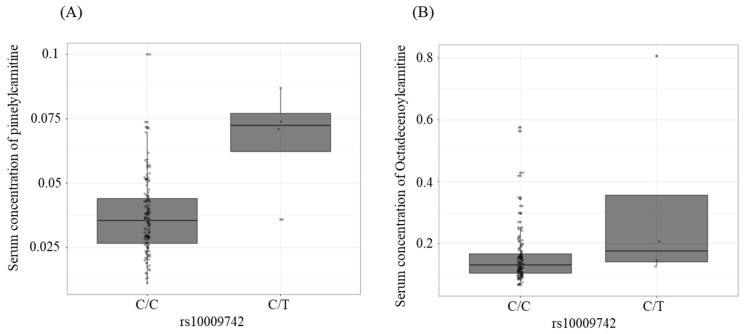
Associations between rs10009742 in the *MARCHF1* gene and serum metabolite concentrations using a linear regression model from GMIA. (**A**) Pimelylcarnitine and (**B**) octadecenoylcarnitine (C: reference allele; T: the effect allele of rs10009742). GMIA, genome–metabolomics integrative analysis; *MARCHF1*, membrane-associated ring-CH-type finger 1.

**Figure 3 metabolites-12-01139-f003:**
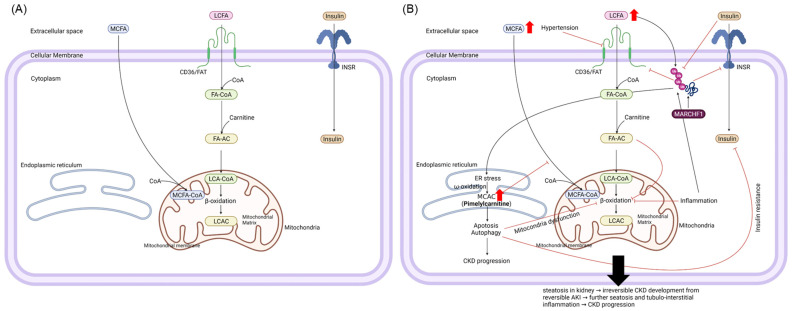
Biological mechanisms for the association between *MARCHF1* and pimelylcarnitine are based on GMIA. An unusual metabolite, pimelylcarintine, may be an intermediator for CKD progression through oxidative stress on the endoplasmic reticulum. (**A**) Normal processes and (**B**) processes during CKD development influenced by the genetic variant of *MARCHF1*, hypertension, and diabetes, and in CKD progression. AC, acylcarnitine; CKD, chronic kidney disease; ER, endoplasmic reticulum; FA, fatty acid; GMIA, genome–metabolomics integrative analysis; INSR, insulin receptor; LC, long-chain; *MARCHF1*, Membrane-associated ring-CH-type finger 1; MC, medium-chain; Ub, ubiquitin. Figures created with BioRender.com.

**Table 1 metabolites-12-01139-t001:** General characteristics of the participants in the discovery and the validation cohorts.

Characteristics	Discovery ^a^ (N = 115)	Validation ^b^ (N = 69)	*p*-Value
	Mean (SD)	Mean (SD)	
Age at baseline	57.2 (8.5)	52.9 (11.9)	0.01
Systolic BP (mmHg)	121.9 (10.6)	134.0 (21.4)	<0.01
Diastolic BP (mmHg)	73.6 (7.4)	78.9 (12.1)	<0.01
Body mass index (kg/m^2^)	24.2 (2.7)	25.5 (3.3)	<0.01
Hemoglobin (g/dL)	13.8 (1.7)	11.7 (1.9)	<0.01
Serum albumin	4.5 (0.3)	3.7 (0.7)	<0.01
eGFR (mL/min/1.73 m^2^)	89.2 (14.5)	53.6 (24.4)	<0.01
eGFR slope (mL/min/1.73 m^2^/year)	−7.1 (2.1)	−6.4 (1.9)	0.05
	**Median [IQR]**	**Median [IQR]**	
Follow-up (years)	3.9 [3.6, 4.2]	1.8 [1.4, 2.4]	<0.01
	**N (%)**	**N (%)**	
Sex (male)	64 (55.7)	49 (71.0)	0.06

Abbreviations: BP, blood pressure; eGFR, estimated glomerular filtration rate; IQR, interquartile range; K-CHIP, the Korean Biobank Array; KNOW-CKD, the Korean cohort study for Outcomes in patients With Chronic Kidney Disease; SD, standard deviation. ^a^. Discovery cohort: K-CHIP consortium. ^b^. Validation cohort: KNOW-CKD cohort.

**Table 2 metabolites-12-01139-t002:** Significant SNPs associated with rapid eGFR decline based on the genome-wide association study.

							K-CHIP Consortium ^1^		KNOW-CKD ^2^	
Chr	Position	SNP	Function	Gene	Allele	MAF	Beta (SE)	*p*-Value	Beta (SE)	*p*-Value
4q32.3	164664101	rs5012631	intronic	*MARCHF1*	C/G	0.03	−4.113 (0.790)	8.69 × 10^−7^	−2.303 (1.118)	0.04
4q32.3	164665383	rs6852270	intronic	*MARCHF1*	C/T	0.03	−4.116 (0.790)	8.55 × 10^−7^	−2.315 (1.118)	0.04
4q32.3	164670375	rs71600637	intronic	*MARCHF1*	C/CAT	0.03	−4.127 (0.790)	8.08 × 10^−7^	−2.377 (1.117)	0.04
4q32.3	164670444	rs1390835129	intronic	*MARCHF1*	CT/C	0.03	−4.129 (0.790)	8.01 × 10^−7^	−2.365 (1.118)	0.04
4q32.3	164674901	rs10009742	intronic	*MARCHF1*	C/T	0.03	−4.128 (0.790)	8.01 × 10^−7^	−2.361 (1.118)	0.04

Abbreviations: Chr, chromosome; eGFR, estimated glomerular filtration rate; K-CHIP, the Korea Biobank Array; KNOW-CKD, The Korean Cohort Study for Outcomes in Patients with Chronic Kidney Disease; MAF, minor allele frequency; SE, standard error; SNP, single-nucleotide polymorphisms. ^1^. Discovery dataset. ^2^. Validation dataset.

**Table 3 metabolites-12-01139-t003:** Genome–metabolomics integrative analysis for rapid eGFR decline.

SNP	Function	Gene	Alleles	MAF	Beta (SE) ^1^	*p*-Value ^1^	Metabolites	Beta (SE) ^2^	*p*-Value ^2^	FDR^2^
rs10009742	intronic	*MARCHF1*	C/T	0.027	−4.128 (0.790)	8.01 × 10^−7^	C7-DC	0.030 (0.007)	7.10 × 10^−5^	1.44 × 10^−2^
rs10009742	intronic	*MARCHF1*	C/T	0.027	−4.128 (0.790)	8.01 × 10^−7^	C18:1	0.167 (0.049)	8.11 × 10^−4^	8.21 × 10^−2^
rs1390835129	intronic	*MARCHF1*	CT/C	0.027	−4.129 (0.790)	8.01 × 10^−7^	C7-DC	0.030 (0.007)	7.10 × 10^−5^	1.44 × 10^−2^
rs1390835129	intronic	*MARCHF1*	CT/C	0.027	−4.129 (0.790)	8.01 × 10^−7^	C18:1	0.167 (0.049)	8.11 × 10^−4^	8.21 × 10^−2^
rs71600637	intronic	*MARCHF1*	C/CAT	0.027	−4.127 (0.790)	8.08 × 10^−7^	C7-DC	0.030 (0.007)	7.10 × 10^−5^	1.44 × 10^−2^
rs71600637	intronic	*MARCHF1*	C/CAT	0.027	−4.127 (0.790)	8.08 × 10^−7^	C18:1	0.167 (0.049)	8.11 × 10^−4^	8.21 × 10^−2^
rs6852270	intronic	*MARCHF1*	C/T	0.027	−4.116 (0.790)	8.55 × 10^−7^	C7-DC	0.030 (0.007)	7.10 × 10^−5^	1.44 × 10^−2^
rs6852270	intronic	*MARCHF1*	C/T	0.027	−4.116 (0.790)	8.55 × 10^−7^	C18:1	0.167 (0.049)	8.11 × 10^−4^	8.21 × 10^−2^

Abbreviations: C7-DC, pimelylcarnitine; C18:1, octadecenoylcarnitine; eGFR, estimated glomerular filtration rate; FDR, false discovery rate; MAF, minor allele frequency; SE, standard error. Gene symbol in the intergenic region was represented by the nearest gene. ^1^. Results of genome-wide association study in the discovery dataset ^2^. Results of genome–metabolomics integrative analysis.

## Data Availability

The K-CHIP consortium genotype data are available upon request under the data sharing policy of the National Research Institute of Health, Korea (https://www.koreanchip.org/blank-8). Other data supporting our findings are available from the corresponding author upon reasonable request.

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
