# Peer review of "Identifying Genetic Variants and Metabolites Associated with Rapid Estimated Glomerular Filtration Rate Decline in Korea Based on Genome–Metabolomic Integrative Analysis"

_metabolites, 2022, doi:10.3390/metabo12111139_

Round 1
Reviewer 1 Report
Comments follow throughout the attached document.

Author Response
We appreciate your kind and detailed comments once again.
We have uploaded an attached file.

Reviewer 2 Report
The article presented a comprehensive summary of a research, the authors aimed to evaluate effect of MARCHF1 which associated with a rapid eGFR decline in patients with hypertension and diabetes. The field of action of the work presents scientific relevance; especially in internal medicine highly associated insulin resistance. The authors concluded that the MARCHF1 gene on chromosome locus 4q32.3 was associated with a very rapid decline in eGFR, an indicator of CKD progression, in patients with hypertension and diabetes in Korea. I am interested in this topic; however, I have several comments:
1. The authors should clarify the feature and novel findings of this study.
2. Previous studies demonstrated that WC and WHtR were both independent risk factor to renal function impairment, which was different to yours study result. Is there other studies which found similar result to yours study?
3. HOMA-IR was found being related to renal impairment; is there any information and further analysis about HOMA-IR in your subjects?
4. The authors should add the comments related to selection bias in this study to the perceived limitation subsection.
5. Please revise the title of your manuscript so that it contains details of the study design which characterize the investigation as well.
6. Due to the mean of eGFR in table showed 89.2 ±14.5 and 53.6 ±24.4, could we infer most of the patients may be in the CKD stage I and II? Did these subjects have other structural or functional abnormality?
7. I am not familiar with GWAS analysis. Please give rationale for this approach.
Author Response

(The authors gave the same response as above.)

Round 2
Reviewer 1 Report
Dear,
The manuscript has been improved according to the indicated indications, and in my opinion it could be published in the Metabolites Journal.
Reviewer 2 Report
Thanks for your great efforts on revision.